# Cohort profile: The Vitality 90+ Study—a cohort study on health and living conditions of the oldest old in Tampere, Finland

Linda Enroth ![ORCID],[1] Pauliina Halonen,[1] Kristina Tiainen,[1] Jani Raitanen,[1,2] Marja Jylhä[1]

¹Faculty of Social Sciences (Health Sciences) and Gerontology Research Center, Tampere University, Tampere, Finland
²The UKK Institute for Health Promotion Research, Tampere, Finland

**Correspondence to**
Dr Linda Enroth;
linda.enroth@tuni.fi

## ABSTRACT

**Purpose** Vitality 90+ is an ongoing population-based study with repeated cross-sectional data collections. The study was designed to examine trends in health, functioning, living conditions, quality of life and care needs among the oldest old in Finland.

**Participants** Nine mailed surveys have been conducted in the city of Tampere between 1995 and 2018. The first three surveys in 1995, 1996 and 1998 included all community-dwelling individuals aged 90 years or older; and the following six surveys in 2001, 2003, 2007, 2010, 2014 and 2018 covered all individuals in Tampere regardless of their living arrangements. In total, the surveys have included 5935 participants (8840 observations). Around 80% of the participants have been women. The participants' age range has been between 90 and 107 years.

**Findings to date** The surveys have consistently asked the same questions over time, covering basic sociodemographic factors, morbidity, functioning, self-rated health (SRH), living arrangements, social relations, quality of life, care needs and providers of care. Survey data have been linked with national register data on health and social service use, mortality and medication. The main findings regarding the time trends show an increase in the proportion of people independent in activities of daily living and mobility. Along with improved functioning, the number of chronic conditions has increased, and SRH has shown a tendency to decline. In addition, we have found increasing occupational class inequalities in functioning and SRH over time.

**Future plans** The next round of data collection will be completed by the end of 2022. The Vitality 90+ Study welcomes research collaborations that fall within the general aims of the project. The research data 1995–2014 are archived at the Finnish Social Science Data Archive and the data for years 2018 and 2022 will be archived in 2023.

## INTRODUCTION

Population ageing has significant social and economic consequences for individuals and societies worldwide. In many countries, the oldest old, that is, people aged over 90 years, are the fastest growing age group.[1] In 1995,

---

### STRENGTHS AND LIMITATIONS OF THIS STUDY

⇒ The major strength of the Vitality 90+ study is total age cohorts of the oldest old including people living in the community and in round-the-clock care.
⇒ The participation rate has been very high (around 80%) in each nine survey rounds from 1995 to 2018.
⇒ National register data on health and social service use, mortality and medication have been linked with the survey data.
⇒ Main limitations of the study are a restricted number of survey questions, and that the data are based on self-reports.
⇒ In oldest old populations, the high mortality rate presents a challenge for individual-based longitudinal analyses.

---

9% of women and 2.5% of men in their birth cohort reached 90 years of age in Finland. By 2018, these figures had risen to 24% for women and 9% for men.[2] This longevity revolution calls for better information not only on the total older population aged 75+ but specifically on the oldest old. Population-based studies on the oldest old are rare, partly because of the challenges involved in research processes. For instance, living in round-the-clock care increases with age and may present difficulties when trying to reach research subjects, and participation in survey studies may be hampered by health conditions such as memory disorders. The Vitality 90+ Study was launched in 1995 to gain a deeper understanding of the situation of very old people.

The Vitality 90+ Study is conducted in the city of Tampere in Finland. It has collected quantitative survey data, register data, functional performance tests, blood samples and qualitative life story interviews with individuals aged 90 years and older (figure 1). The backbone of the study consists of repeated surveys on the population aged 90+, which have been conducted in 1995, 1996, 1998,

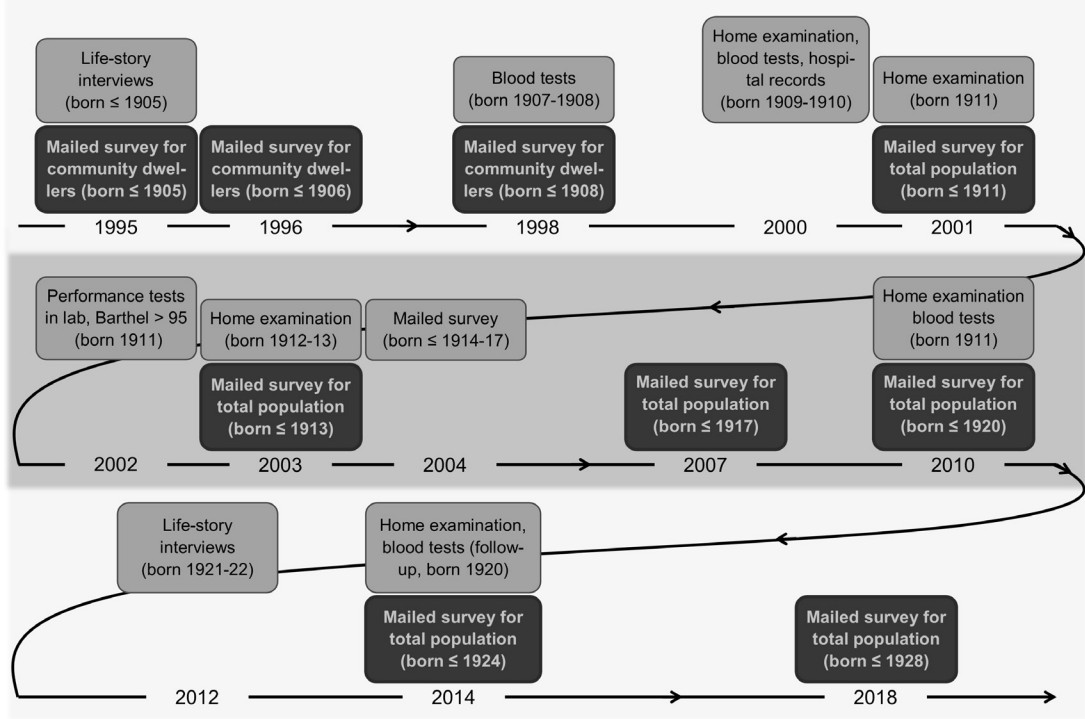

**Figure 1** Data collections in the Vitality 90+ Study from 1995 to 2018.

2001, 2003, 2007, 2010, 2014 and 2018. The main purpose of the study is to examine time trends in health, functioning and quality of life in very old age, but the data also allow for an individual-based longitudinal research design. Survey data linkage with national register data potentiates follow-up studies, such as analyses of predictors of mortality and health and social service use. The Vitality 90+ Study is funded by the Academy of Finland and by competitive grants from several foundations.

The purpose of this cohort profile is to provide a rationale for the Vitality 90+ Study, give an overview of the collected data and summarise the main findings of the project.

## Cohort description

The Vitality 90+ Study is an ongoing cross-sectional population-based survey covering people aged 90 years and older in Tampere, Finland's third largest city with 240 000 inhabitants in 2021. In 1995, 1996 and 1998, a mailed questionnaire was sent to all community-dwelling individuals who were 90 years or older in the study year.[3] In 2001, 2003, 2007, 2010, 2014 and 2018, the questionnaire was mailed to all individuals in Tampere, regardless of their living arrangements. Contact information for the population was obtained from the Tampere city population register. Depending on the funding available, data were collected in 2-to 3-year intervals in the beginning of the study but since 2010, the interval has been set at 4 years. The exact start dates for data collection and response rates are presented in table 1. Questionnaires have mostly been returned within a month (eg, 57% in 2018). Persons not responding within around 6 weeks

have been sent one reminder with a new questionnaire. The next round of data collection will be completed by the end of 2022.

Since all persons aged 90 and older have been invited to participate in each data collection, it has been possible for individuals to participate more than once. The total number of participants in 1995–2018 is 5935, while the number of observations has reached 8840. The number of participants with two observations is 1475 (25%), and 602 (10%) persons have participated in the study at least three times. Most of the participants have been women and the proportion of those living in round-the-clock care is in the range of 30%–40% (2001–2018). The study allows the use of proxy respondents. Responses are considered proxy reported if they are given by a family member, friend or a staff member at home care services or in round-the-clock care, and independent if given by the participant alone or with practical help only, for instance in writing. The data concerning the total population in the area (2001–2018) show that most of the responses have been given by the study participant (77%–87%).

The possible selection bias in the study samples is assessed by examining individual-based short-term mortality, which is derived from the population register and linked with survey data using personal identification codes (PICs). PIC is a unique number provided at birth or on the grounds of staying in Finland, which remains the same throughout the individual's life course. During the first 2 months after each data collection between 2001 and 2018, mortality was higher for those not participating in the study (10%–24%) compared with those who did

**Table 1** Characteristics of the Vitality 90+ Study populations in 1995–2018 by sex separately for respondents and non-respondents

| | Total | | Women | | | | Men | | | | |
| | | | Respondents | | Non-respondents | | Respondents | | Non-respondents | | |
| Study year | N | Response rate | n | Age, mean (SD) | n | Age, mean (SD) | n | Age, mean (SD) | n | Age, mean (SD) | Data collection started |
|---|---|---|---|---|---|---|---|---|---|---|---|
| 1995 | 448 | 81.7 | 300 | 91.9 (2.2) | 62 | 92.5 (2.7) | 66 | 91.9 (1.9) | 20 | 92.9 (2.4) | 5.5.1995 |
| 1996 | 506 | 82.0 | 338 | 91.8 (2.1) | 74 | 92.3 (2.6) | 77 | 92.0 (2.1) | 17 | 91.9 (2.3) | 20.9.1996 |
| 1998 | 556 | 84.5 | 363 | 92.0 (2.1) | 67 | 91.9 (1.8) | 107 | 91.9 (2.1) | 18 | 91.7 (1.5) | 15.11.1998 |
| 2001 | 1063 | 83.9 | 720 | 92.4 (2.6) | 139 | 92.3 (2.4) | 172 | 92.0 (2.3) | 32 | 91.9 (2.2) | 15.5.2001 |
| 2003 | 1113 | 86.3 | 771 | 92.5 (2.6) | 130 | 92.6 (2.5) | 190 | 91.9 (2.0) | 22 | 91.7 (2.2) | 21.5.2003 |
| 2007 | 1147 | 82.3 | 751 | 92.7 (2.6) | 168 | 92.8 (2.3) | 193 | 92.2 (2.2) | 35 | 92.4 (1.8) | 12.12.2007 |
| 2010 | 1606 | 79.5 | 1037 | 92.7 (2.7) | 273 | 92.7 (2.7) | 240 | 92.2 (2.3) | 56 | 92.4 (2.4) | 25.2.2010 |
| 2014 | 2056 | 79.6 | 1259 | 92.8 (2.7) | 330 | 93.0 (2.8) | 378 | 92.2 (2.2) | 89 | 92.2 (2.3) | 16.1.2014 |
| 2018 | 2449 | 76.7 | 1387 | 92.8 (2.7) | 464 | 92.9 (2.8) | 491 | 92.3 (2.5) | 107 | 92.4 (2.0) | 26.3.2018 |

participate (1%–2%). Thus, it is likely that the sickest and most disabled individuals are missing from our study, as tends to be the case in survey studies that focus on older people.

The demographic characteristics of the city of Tampere, a doubling of the 90+ population, an increase in the relative proportion of the 90+population and the sex distribution, are all in line with the overall demographic changes in Finland between 2001 and 2018.[4] The oldest old populations in both Finland and Tampere are rather homogenous in terms of ethnic background and language.[5]

Furthermore, Tampere comprises both urban and rural areas and has a mix of both affluent and disadvantaged areas. In our understanding, Vitality 90+ is highly representative of the 90+population in Finland.

## Follow-ups and measurements
The Vitality 90+ Study comprises nine repeated cross-sectional data collections. Figure 2 shows the number of respondents and non-respondents, participation in the study by birth cohort and mortality for each round of data collection. The response rate has been high (around

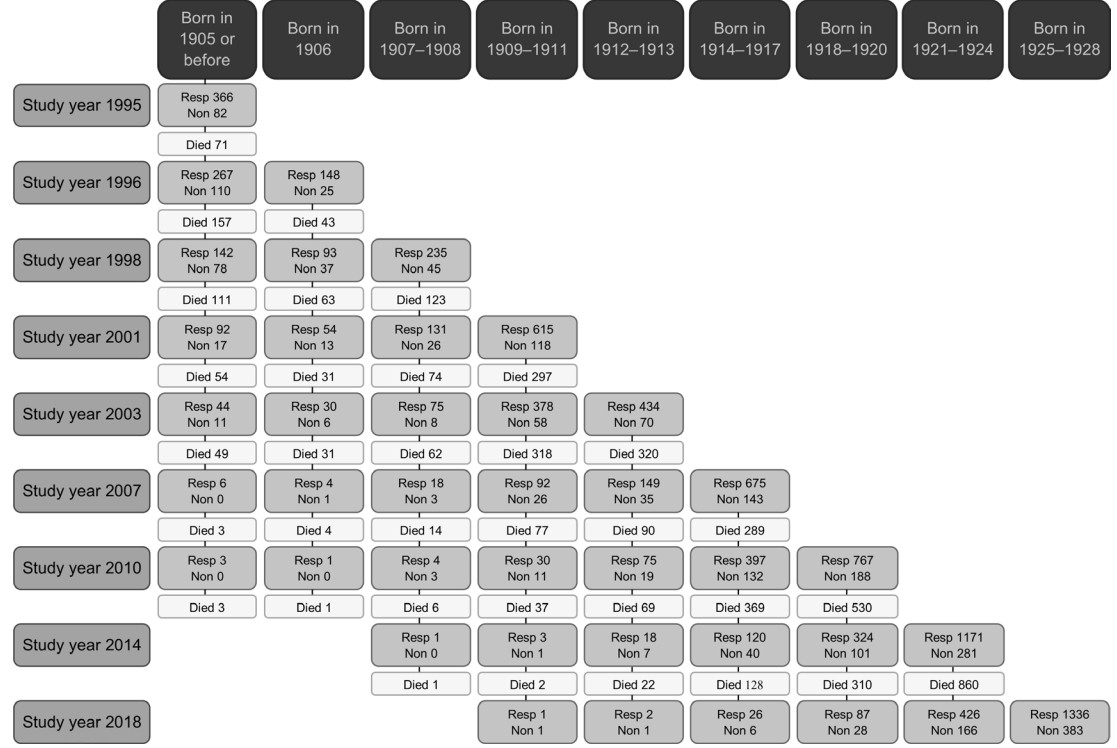

**Figure 2** Number of respondents (Resp), non-respondents (Non) and those who died in the Vitality 90+ Study between 1995 and 2018. Rows represent study years and participation by birth cohorts. Columns represent birth cohorts, their mortality and participation in particular study years.

**Table 2** Overview of content in the Vitality 90+ Study between 1995 and 2018

| Survey data | |
| --- | --- |
| **Theme of questions** | **Items** |
| Basic information | Age, sex, respondent (with help or proxy), main life-time profession, highest attained level of education (2010) |
| Physical environment | Place of living, place of stay at the time of answering, preferred place of residence in current situation (2014) |
| Social environment | Marital status (2010), household members, contacts with family members, relatives or friends (2001) |
| Service environment and need of help | Help by care workers, help at home in everyday life, informal care |
| Health conditions | Self-rated functioning (2010), self-rated health |
| Self-reported chronic conditions diagnosed by a doctor | Depression, hip fracture, hypertension, heart diseases, cancer, dementia (Alzheimer's disease, other dementia or worsening of memory), stroke, diabetes, arthrosis, Parkinson's disease, cerebrovascular disorder, calcification of blood vessels, inflammatory arthritis, chronic lung disease (2018) |
| Physical functioning | Mobility and activities of daily living |
| Activity limitations and participation restrictions | Frequency of moving outdoors<br>Device use when moving outdoors (2001) |
| Impairments and symptoms | Ability to read newspaper (2010), ability to hear (2010)<br>Dizziness (2014), poor balance (2014), fatigue (2014), pain (2018) |
| Subjective experiences | Life satisfaction (2014), feeling lonely (2018), status of older people |
| **Register data** | |
| **Register** | **Items** |
| Finnish Institute for Health and Welfare | Care registers for healthcare: home care, primary care and specialised care |
| Social Insurance Institution of Finland | Prescription database of drug purchases |
| Statistics Finland | Causes of death register: causes of death |
| Digital and Population Data Services Agency | Population register: date and place of death |

Note: since the start of the study in 1995, new questions have been included in the questionnaire. The year in the brackets indicates when the item was included in the study for the first time.

80%) in each survey round. However, most of the study participants have only one observation. That is due to high mortality, especially between later study rounds with 4 years interval. For instance, of the study participants in 1995, 12% died, 19% were eligible but did not respond in 1996 and 69% responded in 1996. The corresponding figures for the new participants in 2010 were as follows: 52% died, 10% were eligible but did not respond in 2014 and 38% responded in 2014.

As a rule, the questionnaires have been carefully completed. The share of observations with missing information on the main outcomes of the study—health, functioning and quality of life—has been <5%. The items include questions on basic sociodemographic factors, morbidity, activities of daily living (ADL) and mobility, self-rated health (SRH), symptoms (eg, pain, dizziness), living arrangements, social relations, quality of life, care needs and providers of care (table 2). Throughout the study years, the data collection mode and the questions regarding the main outcomes have been identical. However, some new items have been included in later study rounds.

National administrative register data have been linked with the survey data using PICs. This allows us to examine the use of health and social care services (eg, admissions to round-the-clock care), predictors of mortality and medication use. In addition, information on dates of death has been used to assess selection bias between the study respondents and non-respondents.

Data on home care, primary care, specialised care and round-the-clock care are obtained from care registers administered by the Finnish Institute for Health and Welfare. Information on date and place of death is obtained from the population register and on causes of death from Statistics Finland. Information on prescription drug purchases is provided by The Social Insurance Institution of Finland. Register data are updated regularly.

Study protocols have been approved by the Ethics Committee of the City of Tampere and, for more recent years, by the Regional Ethics Committee of Tampere University Hospital. The study has a research permit from the city of Tampere, and all participants or their representatives have provided written informed consent.

The Vitality 90+ Study has also collected data from smaller subgroups (figure 1). Data from anthropometric measurements and functional performance tests were collected in 2000, 2001, 2002, 2003, 2010 and 2014 among persons aged 90–91 years.[6 7] Blood samples were taken in 1998, 2000, 2010 and 2014 for biological examinations, which have focused on ageing of the immune system and epigenetics.[8–11] Face-to-face life story interviews were collected in 1995 (n=250) and 2012 (n=45). Qualitative interviews have been concerned with subjective experiences of long life, for instance perceptions of social relationships[12] and successful ageing.[13]

### Patient and public involvement
Study population or the general public were not involved in the planning, design or conduct of the study. The city of Tampere supported the recruitment of the study participants and the data collections have been advertised in local newspapers and social media.

### Findings to date
The Vitality 90+ Study is a unique data source on the oldest old, comprising nine repeated cross-sectional data collections over a period of 23 years. The main research focus has been on the health consequences of lengthening old age,[4 5 14] predictors of mortality[15–17] and round-the-clock care.[15 18–20] Other themes include social inequality in health[18 21 22] and the concepts of successful ageing[17 19 23] and SRH.[5 22 24 25] Vitality 90+ is an excellent data source for examining time trends, that is, changes in health and functioning over time in the 90+ population. However, also individual-based longitudinal analyses on changes in functioning and SRH have been conducted.[26–28]

### Descriptive findings of the Vitality 90+ Study
Women live longer than men and therefore account for the majority of the 90+ population. From 2001 to 2018, the proportion of birth cohorts reaching 90 years grew more rapidly among men than women, and the proportion of men in the 90+ population increased from 19% to 26%. Most of the study participants are widows, more than one-third live in round-the-clock care, and more than half of community-dwelling men and more than 80% of community-dwelling women live alone. In this age group, the most important source of help with daily activities is from children and grandchildren with their families. In addition, 26% have frequent formal home care visits. Despite the high number of chronic conditions and high prevalence of disabilities, almost 90% of the study respondents have indicated that they are rather or very satisfied with their current life.

### The concepts of successful ageing and SRH
The Vitality 90+ Study has assessed successful ageing among the long-lived individuals using multidimensional models that include physical, social and psychological components. The prevalence of successful ageing ranges from 1.6% to 18.3% depending on how the concept is defined.[23] The most demanding criteria that led to the lowest prevalence of successful ageing were characterised as the absence of diseases, good functioning, psychological well-being and having social contacts. The highest prevalence of successful ageing was met with the criteria allowing other diseases but not dementia, good functioning and having social contacts. Sociodemographic factors such as younger age, male gender and higher level of education are associated with the models of successful ageing.[23]

SRH is a comprehensive measure of health with strong predictive value for future health events. In Vitality 90+, one-third of the study participants assess their health as good or excellent. The main factors associated with poor SRH are fatigue, depression, difficulties in mobility, dizziness, deficits in vision and heart disease.[24]

### Time trends of health and functioning
Public health policies are aimed at helping people to live longer lives with more healthy life years. Old age mortality has indeed declined sharply, but less is known about the development of health and functioning among the oldest old. Our results show that in 2018, a higher proportion of people aged 90+ are independent in ADL and mobility than 17 years earlier.[4] Combining information on mortality and functioning, our analysis showed that of the remaining life expectancy at the age of 90, the length of time lived without difficulties in ADL and mobility increased from 2001 to 2018. However, the length of time lived with difficulties in ADL and mobility did not decrease. Therefore, at the population level, the improvement in functioning was not enough to offset the increase in care needs.[4] With regard to self-reported doctor-diagnosed chronic conditions, the prevalence of diabetes and hypertension has increased over time. The prevalence of dementia is high in this age group, standing at over 40% throughout this study. Recent results show that along with the tendency for the prevalence of dementia to decline, the number of comorbidities with dementia has increased over time.[29] Figure 3 illustrates the trends in functional ability and multimorbidity. Along with improved functioning and an increasing number of chronic conditions, good SRH has shown a tendency to decline.[5] The findings on improved functioning are largely driven by favourable trends in higher occupational classes, while the decline in SRH is due to adverse trends in lower occupational classes.[22] Our results show increasing occupational class inequalities in functioning and SRH from 2001 to 2018.[22]

### Longitudinal changes in health and functioning
An individual-based longitudinal analysis of changes in physical functioning among the oldest old showed significant age-related decline in functioning, which accelerated with age.[26] Following the same people over time, Vitality 90+ has shown that even though men have better functioning than women, the rate of decline did not differ between the sexes over 2–9 years of follow-up. Among the youngest study participants, however, those

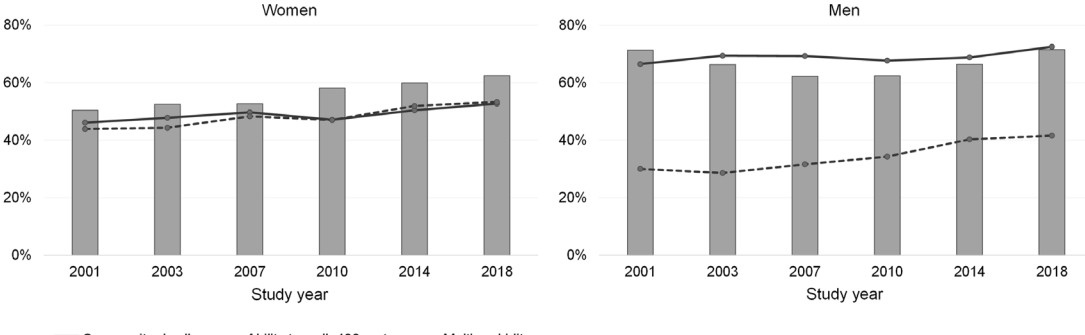

**Figure 3** Prevalence of the ability to walk 400 metres (without difficulty), multimorbidity (at least three chronic conditions) and the proportion of community-dwellers.

with more chronic conditions showed a steeper decline in functioning.[28] Another longitudinal analysis showed that while women live longer than men, they spend more time with disability and multimorbidity.[14] Furthermore, Vitality 90+ has published results on longitudinal changes in SRH. The study suggests a decline in average SRH as individuals grow older and that the changes are sensitive to increases in the number of chronic conditions and decline in functioning.[25]

### Use and predictors of round-the-clock care

Current social policies encourage living at home for as long as possible. As a consequence, round-the-clock care coverage has decreased in Finland. From 2001 to 2018, the proportion of Vitality 90+ participants living in round-the-clock care fell from 40% to 30%. The most important predictor for entering round-the-clock care is dementia.[15] Other identified predictors in the 90+ population are multimorbidity, that is, having at least two chronic conditions, living alone and receiving help at home with daily activities. Women have a higher risk for entering round-the-clock care than men, which is partly related to the fact that a larger proportion of women live alone.[20] Furthermore, individuals characterised as successful agers tend to have a lower incidence of entering round-the-clock care.[19] As the number of the oldest old is set to continue to rise and as this age group has the highest need for round-the-clock care, it is essential that we properly understand the predictors and the needs of the people entering this type of care.

### Predictors of mortality

Mortality is high among nonagenarians. In 2018, 16% of 90-year-olds in Finland died within 1 year.[2] Our follow-ups from 3 to 9 years show that by and large, the same factors—older age, male gender, functional disability, poor SRH and lower occupational class—predict mortality in the oldest old population as in younger old populations.[16 27] We have also found that severe multimorbidity and chronic conditions such as dementia and heart disease are risk factors for mortality.[15] In addition, successful ageing is a predictor of longer life in nonagenarians.[17]

### Strengths and limitations

The main strengths of the Vitality 90+ Study are total age cohorts of the oldest old that include people living in the community and in round-the-clock care and the high participation rates (around 80%). The study also encourages the use of proxy respondents to reach people with memory disorders and disabilities. Throughout the study years, the data collection mode and the questions regarding the main outcomes have been identical. This provides an excellent data source for examining time trends. Furthermore, the data collected have been linked with exhaustive register data on health and social service use, prescription drug purchases and mortality. Additionally, with access to information on age, sex and mortality for non-respondents, we have been able to assess mortality selection.

The main limitation of the study is the restricted number of survey questions, a conscious choice we have made to guarantee a high response rate. In addition, there are some other weaknesses that are common to all population-based studies and studies concerned with the oldest old. First, collecting data in a large population generally means having to rely on self-reported information. Second, in oldest old populations, the high mortality rate presents a challenge for individual-based longitudinal analyses. Furthermore, even though the contact information and vital status are received just before sending out the questionnaires, a small proportion of the target population has passed away before receiving the questionnaire.

### COLLABORATION

The study is conducted in collaboration with experts from different fields (health sciences, sociology, biology, demography) and across national borders.[8 14 22] The Vitality 90+ Study welcomes research collaborations that fall within the general aims of the project. We follow the EU general data protection regulation (679/2016) and the Finnish Data Protection Act (1050/2018) in data management and use. The metadata for the study are archived at the Finnish Social Science Data Archive (https://services.fsd.tuni.fi/catalogue/series/64?lang=

en). For additional details, contact the principal investigator of the study Linda Enroth.

**Acknowledgements** The Vitality 90+ Study was initiated by Marja Jylhä and Antti Hervonen. We wish to thank all the researchers who have contributed to the Vitality 90+ Study since 1995. We also wish to thank the city of Tampere and the elderly care units in Tampere for their help in data collection.

**Contributors** LE and MJ initiated the manuscript. LE, PH, KT, JR and MJ contributed to planning and writing the manuscript. LE drafted the first version of the manuscript and JR designed and prepared the data flowcharts. All authors reviewed and contributed to the manuscript and approved the final version. All authors are accountable for all aspects of the work. LE is responsible for the overall content as the guarantor.

**Funding** This work was supported by the Academy of Finland through its funding to the Centre of Excellence in Research of Ageing and Care (CoEAgeCare, grants numbers 326567 and 336670).

**Competing interests** None declared.

**Patient and public involvement** Patients and/or the public were not involved in the design, or conduct, or reporting or dissemination plans of this research.

**Patient consent for publication** Not applicable.

**Ethics approval** The study protocol has been approved by the Ethics Committee of the city of Tampere and, for more recent years, by the Regional Ethics Committee of Tampere University Hospital (R18041). Participants gave informed consent to participate in the study before taking part.

**Provenance and peer review** Not commissioned; externally peer reviewed.

**Data availability statement** Data are available upon reasonable request. Vitality 90+ datasets are described in a public, open access repository (Finnish Social Science Data Archive) and data are available upon a reasonable request.

**ORCID iD**
Linda Enroth http://orcid.org/0000-0001-6613-3154

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
