## [Reviewer comments · BMJ Open]

ARTICLE DETAILS

TITLE (PROVISIONAL)	Cohort profile: The Vitality 90+ Study. A cohort study on health and living conditions of the oldest old in Tampere, Finland
AUTHORS	Enroth, Linda; Halonen, Pauliina; Tiainen, Kristina; Raitanen, Jani; Jylhä, M

VERSION 1 – REVIEW

REVIEWER	Liljas, Ann Karolinska Institutet, Global Public Health
REVIEW RETURNED	10-Oct-2022

GENERAL COMMENTS	Thank you for the opportunity to read and comment on this well-written manuscript of the Vitality 90+ cohort profile. Please see my comments below. Abstract: Participants: Replace 'their place of living' (as it makes the reader wonder whether they could live outside Tampere) with 'living arrangements'. Future plans: Consider adding 'in the end of' in the sentence "will be carried out in the end of 2022" or replace it with something like "are currently collected (2022)" as it to some extent confuses the reader as 2022 is soon coming to an end. Introduction, page 3, the purpose of the Vitality 90+ Study is clearly expressed however the purpose of this paper/document is not outlined. Therefore, it would be useful if this section ends by a study rationale of 1-2 sentences explaining why you have put together this overview of the study e.g. there is no previous document that describes the cohort Cohort description, page 4, line17, to be consistent with the text in the Abstract, replace 'their place of living' with 'living arrangements' Cohort description, page 4, line26, same as in the abstract about Future plans, please be more specific when in 2022 data will be collected Page 6, line 10, "PIC is a unique number identifying each individual..." would it be possible (and correct?) to add "PIC is a unique number provided at birth or when gaining citizenship identifying each individual..." Page 6 following/part of the paragraph on demographic characteristics of Tampere, it would be useful if you could add whether Tampere is 'about average' of Finnish cities/societies and has a mix of both affluent and disadvantaged areas or more of any
--

	of these two compared to other cities and areas of Finland. Page 10, line 38-40 Please refer the reader to where to find out about these different definitions of successful ageing Page 10, line 40, and page 13 line 10, consider replacing 'male sex' with 'being male' Page 11, line 28, if the prevalence of dementia was doctor-diagnosed, this should be specified: "The prevalence of doctor-diagnosed dementia..." Limitations, page 13, line 35, it would be useful if you provide an example of these exceptions Limitations, page 13, line 49, 'research questions' – should it be just questions or survey questions? In the Discussion or Limitations, it would be useful if you comment on why the response rate has gone down. Such comment could refer to similar trends in other cohorts of the older people. In the Discussion or Limitations, if possible, add any information on what is done to avoid that surveys are sent out someone who recently died. In some cohorts of older adults, the survey is sent out just days after the death register has been updated to minimize the risk of sending the survey to someone who has died, as close family members have complained about having to pick up post sent to their dead relative.
--	--

REVIEWER	Seematter-Bagnoud , Laurence Center for Primary Care and Public Health , Epidemiology and Health systems
REVIEW RETURNED	06-Dec-2022

GENERAL COMMENTS	This is a very well written manuscript, that thoroughly describes the Vitality90+ study. I only have some minor comments (more precisely, requests for additional information): 1) You never explain why the intervals between surveys are irregular (financing opportunities/difficulties would be my guess). Although it is not really a methodological limitation, regular intervals would be an advantage when examining trends, and would be easier to summarize and present to the audience. 2) I was surprised that only about a third of participants had 2+ observations (25% participated twice and 10% participated 3+ times, see page 6), although the interval between some assessment was not very long (e.g. 2 years). The justification for this is the high mortality rate, and you refer to Figure 2 but do not provide any figure for this at page 6. Figure 2 is rather dense to read, so I would suggest to add some estimates for mortality rates at page 6, as well as some estimates of non-response among those still eligible to participate. The reader gets the information at page 13 that 16% of the sample died within one year in 2018. 3) Page 10, paragraph successful ageing: You mention that the prevalence of successful ageing ranges from 1.6% to 18.3%
--

	depending on the definition used. I guess that the information stems from reference 23, which appears in the next sentence. However, as a reader, it is somehow frustrating to get such a wide range of prevalence without any other information: perhaps you could mention which definition led to the lower estimate of 1.6%, and which other definition led to the highest estimate, or add any other relevant information.
--	---

VERSION 1 – AUTHOR RESPONSE

Reviewer: 1

Dr. Ann Liljas, Karolinska Institutet

Comments to the Author:

Thank you for the opportunity to read and comment on this well-written manuscript of the Vitality 90+ cohort profile. Please see my comments below.

1. Abstract: Participants: Replace ‘their place of living’ (as it makes the reader wonder whether they could live outside Tampere) with ‘living arrangements’.

Response: Thank you for the suggestion. We changed the wording as suggested.

2. Future plans: Consider adding ‘in the end of’ in the sentence “will be carried out in the end of 2022” or replace it with something like “are currently collected (2022)” as it to some extent confuses the reader as 2022 is soon coming to an end.

Response: Thank you for the comment. The manuscript was submitted in September when the situation with data collection was very different. We have reframed the sentence: The next round of data collection will be completed by the end of 2022.

3. Introduction, page 3, the purpose of the Vitality 90+ Study is clearly expressed however the purpose of this paper/document is not outlined. Therefore, it would be useful if this section ends by a study rationale of 1-2 sentences explaining why you have put together this overview of the study e.g. there is no previous document that describes the cohort

Response: We have added the rationale of this article to the end of the introduction.

“The purpose of this cohort profile is to provide a rationale for the Vitality 90+ Study, give an overview of the collected data and summarize the main findings of the project.”

4. Cohort description, page 4, line17, to be consistent with the text in the Abstract, replace ‘their place of living’ with ‘living arrangements’

Response: Edited as suggested.

5. Cohort description, page 4, line26, same as in the abstract about Future plans, please be more specific when in 2022 data will be collected

Response: Edited similarly as presented in point 2.

6. Page 6, line 10, "PIC is a unique number identifying each individual..." would it be possible (and correct?) to add "PIC is a unique number provided at birth or when gaining citizenship identifying each individual..."

Response: Thank you for the question. In Finland, the PIC is, in most cases, provided at birth. However, foreign-born people do not need a citizenship in order to get PIC. For instance, students and people with residence permit can apply for a PIC. See more on <https://dvv.fi/en/foreigner-registration/>

We have revised the part as follows: PIC is a unique number provided at birth or on the grounds of staying in Finland, which remains the same throughout the individual's life course.

7. Page 6 following/part of the paragraph on demographic characteristics of Tampere, it would be useful if you could add whether Tampere is 'about average' of Finnish cities/societies and has a mix of both affluent and disadvantaged areas or more of any of these two compared to other cities and areas of Finland.

Response: Thank you for the suggestion. Tampere has both affluent and disadvantaged areas and based on Kurvinen and Sorri 2016, socioeconomic segregation is about the average of the capital area and the city of Turku, which is the sixth populated city in Finland. In the study, socioeconomic segregation was characterized with median income, proportion of employed persons and proportion of employed persons with high level of education. Belonging to the lowest quintile in all three measures indicated segregation in the area (250 m x 250 m). Unfortunately, the article is in Finnish. <https://journal.fi/janus/article/view/60254>

We revised the paragraph on demographic characteristics: "Furthermore, Tampere comprises both urban and rural areas and has a mix of both affluent and disadvantaged areas."

8. Page 10, line 38-40 Please refer the reader to where to find out about these different definitions of successful ageing

Response: As the other reviewer asked for clarification for the successful ageing definitions, we have revised that part. The reference is placed right after the sentence explaining the differences in the prevalence of successful ageing.

9. Page 10, line 40, and page 13 line 10, consider replacing 'male sex' with 'being male'

Response: Thank you. We changed the wording as male gender.

10. Page 11, line 28, if the prevalence of dementia was doctor-diagnosed, this should be specified: "The prevalence of doctor-diagnosed dementia..."

Response: Thank you for the comment. As all diseases are self-reported doctor-diagnosed conditions, we have added that information into table 2. We have also reframed the

sentence on page 12 “With regard to self-reported doctor-diagnosed chronic conditions, the prevalence of diabetes and hypertension has increased over time.”

11. Limitations, page 13, line 35, it would be useful if you provide an example of these exceptions

Response: Thank you for the suggestion. Actually, we have had the same mode of data collection and identical questions for the main outcomes throughout the study years. However, we have added new questions to the questionnaire in later study years. We have revised two parts of the manuscript on pages 7 and 14.

We also added this information into table 2.

Note: Since the start of the study in 1995, new questions have been included in the questionnaire. The year in the brackets indicates when the item was included in the study for the first time.

12. Limitations, page 13, line 49, ‘research questions’ – should it be just questions or survey questions?

Response: Thank you for noticing the mistake. Yes, we mean survey questions. We have revised the text as suggested.

13. In the Discussion or Limitations, it would be useful if you comment on why the response rate has gone down. Such comment could refer to similar trends in other cohorts of the older people.

Response: Thank you for the comment. There is some variation in the response rate, and it has been a little lower in the later study years. It is good to remember that in the first three study rounds, study population was based on community-dwelling individuals and from 2001 onwards, all individuals, also those living in round-the-clock care were included. We would expect somewhat higher response rate for the first study rounds because the healthier end of the population was examined. However, as to our understanding, the response rate is exceptionally high in each survey round, especially with total age cohorts of the oldest old.

We agree that the response rates are declining also among older populations. It may be related to increasing requests to participate in different studies, marketing surveys and political polls, burdensome long questionnaires with lengthy consent forms, increasing interest in data protection issues and benefits of the study and so on.

Ref <https://doi.org/10.1016/j.annepidem.2007.03.013>

Unfortunately, we do not know what the driving force for the decline in response rate in this study is. We have seen a decline in proxy responses over the years, which could imply increasing health selection. We have tried to assess selection by examining individual-based short-term mortality (page 5). It seems that mortality was clearly higher for those who did not participate in the study when compared to those who participated in the study. In our sensitivity analysis, we did not find differences in mortality selection in later study years (2010, 2014 and 2018). Thus, we do not have any evidence that, for instance in 2018, there would be more non-respondents among people with poor health than in 2010.

14. In the Discussion or Limitations, if possible, add any information on what is done to avoid that surveys are sent out someone who recently died. In some cohorts of older adults, the survey is sent out just days after the death register has been updated to minimize the risk of sending the survey to someone who has died, as close family members have complained about having to pick up post sent to their dead relative.

Response: For each data collection, we have requested information of the study population (names and addresses) from the city of Tampere. The Tampere city population register has a couple of weeks delay in mortality updates, and it takes a couple of weeks to print and mail the questionnaires. Even though the study process is as fast as possible, some people have died before the questionnaire is delivered to their place of living. That is very unfortunate but unavoidable in a survey study. The information material sent out includes our contact information, and some people have contacted us to tell that they have just lost their spouse or a parent.

We added a sentence to the limitations. Furthermore, even though the contact information and vital status are received just before sending out the questionnaires, a small proportion of the target population has passed away before receiving the questionnaire”

VERSION 2 – REVIEW

REVIEWER	Liljas, Ann Karolinska Institutet, Global Public Health
REVIEW RETURNED	27-Dec-2022
GENERAL COMMENTS	Thank you for your effort and time updating the manuscript. I have no further suggestions/comments.